# Effectiveness of CKD Exacerbation Countermeasures in Izumo City

**DOI:** 10.3390/jpm11111104

**Published:** 2021-10-28

**Authors:** Takafumi Ito, Fumika Kamei, Hirotaka Sonoda, Masafumi Oba, Miharu Kawanishi, Ryuichi Yoshimura, Shohei Fukunaga, Masahiro Egawa

**Affiliations:** Division of Nephrology, Shimane University Hospital, 89-1 Enyacho, Izumo 693-8501, Shimane, Japan; kmifmk0524@gmail.com (F.K.); hirotakasonoda@yahoo.co.jp (H.S.); m.oba117@med.shimane-u.ac.jp (M.O.); miharu.1709@gmail.com (M.K.); ryoshimura110@gmail.com (R.Y.); fukunaga@med.shimane-u.ac.jp (S.F.); koro211@med.shimane-u.ac.jp (M.E.)

**Keywords:** chronic kidney disease, Kidney Disease Countermeasures Study Group Report, specific health checkup, medical care system, CKD exacerbation

## Abstract

To diagnose chronic kidney disease (CKD) at an early stage, it is important to promote appropriate health guidance and consultation recommendations through regular medical examinations and implementation of continuous high-quality and appropriate treatment. From fiscal year (FY) 2018, Izumo City has initiated the “Izumo City CKD Exacerbation Countermeasures” program. In this study, we aimed to report on the methods undertaken and the effects of this program. Residents aged 40–74 years who underwent specific health checkups from the Izumo City National Health Insurance in FY2018 and FY2019 were included. The rates of CKD re-examination candidates, re-examinations implementation, nephrologist referrals, and health guidance referrals between FY2018 and FY2019 were compared. The rate of CKD re-examination candidates in both years remained unchanged at approximately 7%. The rate of re-examination implementation in FY2019 significantly increased relative to that in FY2018 (*p* < 0.001). Subsequent re-examination candidate trends showed that the rate of nephrologist referrals did not increase. However, the rate of city health guidance referrals significantly increased (*p* < 0.001). Increase in the re-examination and health guidance examination rates indicate improved awareness of CKD among the public and family doctors, and it is expected to prevent CKD exacerbation in the future.

## 1. Introduction

Chronic kidney disease (CKD) in Japan tends to increase annually alongside lifestyle changes and aging; by the end of 2019 ~350,000 patients had been undergoing dialysis [1]. Furthermore, CKD is closely related to cardiovascular disease [2] and cognitive dysfunction [3], and it has been attracting attention as a new national disease that adversely affects the healthy life expectancy of Japanese citizens. Therefore, in July 2018, the Ministry of Health, Labor, and Welfare issued the “Kidney Disease Countermeasures Study Group Report” that proposed five individual countermeasures against kidney disease [4,5]: “dissemination and public awareness,” “improvement of local medical care provision systems,” “improvement of medical care standards,” “human resource development,” and “promotion of research and development.” Among these, “improvement of local medical care provision systems” aims to create a medical care cooperation system for early detection and diagnosis of CKD, as well as implementation of continuous high-quality and appropriate treatment. This can be achieved with the cooperation of medical staff; promotion of coordination between family doctors, nephrology specialized institutions through referrals/counter referrals and a two-person doctor system; and wide dissemination of referral criteria from family doctors to nephrology specialized institutions. In addition, appropriate health guidance and encouragement are promoted through regular health examinations. All these efforts have proven to be effective [6].

Izumo City is the second most populous city in Shimane Prefecture, with a population of 174,812 inhabitants as at 31 July 2021, and an aging rate of 29.92% for the individuals aged >66 years. There are two nephrology specialized institutions, 11 nephrologists, two diabetes specialized institutions, and 19 diabetologists. The search for countermeasures for diabetic nephropathy exacerbation has been ongoing for some time in the Izumo City; nephrologists have started participating in these efforts and several discussions on how to prevent CKD exacerbation have been held. 

The “Izumo City CKD Exacerbation Prevention Countermeasures” program using specific health checkups (Figure 1) was initiated in 2018. According to the Clinical Practice Guidebook for Diagnosis and Treatment of Chronic Kidney Disease 2012 [7], which is followed in Japan, patients with an estimated glomerular filtration rate (eGFR) level <50 mL/min should be referred to a nephrologist; we used this criterion for patient selection in this program. Herein, we aim to report the methods undertaken and the effects of this program on the public.

## 2. Methods

Subjects aged 40–74 years who underwent the Izumo City National Health Insurance specific health checkup in the fiscal year (FY) 2018 (11,685 people) and FY2019 (11,315 people) were included in this study. There were 869 individuals who met the re-examination criteria in FY2018 and 906 in FY2019, all of whom were included in the analysis. We sent for “re-examination, specific health checkup (guidance)” (Figure 2), and recommend ed a visit to a family doctor to the subjects who fulfilled any of the following criteria: eGFR < 50 mL/min/1.73 m^2^ (eGFR < 40 mL/min/1.73 m^2^ for those aged > 70 years); serum creatinine (Cr) > 1.01 mg/dL for males and > 0.9 mg/dL for females, or urinary protein > 2+. The family doctors were instructed to perform a urinalysis (qualitative urine measurements, urinary protein/Cr ratio) and serum Cr and eGFR re-examinations for patients who brought referral letters. The doctors were requested to refer the patients to a nephrologist if they satisfied any of the following: (1) urinary protein/Cr ratio of > 0.5 g/gCr or urinary protein of > 2+; (2) eGFR of < 50 mL/min/1.73 m^2^ (eGFR < 40 mL/min/1.73 m^2^ for those aged > 70 years); or (3) positive for both urinary protein and occult blood (i.e., ≥1+) (Figure 3). Additionally, we instructed the family doctors to refer those who did not meet the above criteria to the Izumo City’s “Well-Being Health Counseling” (health guidance by public health nurses, dietitians, and health exercise instructors) as needed. Furthermore, nephrologists provided additional explanations on the implementation of CKD exacerbation countermeasures in briefing sessions for doctors regarding special health examinations and special health guidance prior to specific health checkups.

The eGFR was calculated at each health checkup facility using a three-variable equation modified for Japanese patients [8]: eGFR (mL/min/1.73 m^2^) = 194 × age (year)^−0.287^ × serum creatinine (mg/dL)^−1.094^ × 0.739 (if female). 

In Japan, the specific health checkup includes urine qualitative analysis as one of the required items, but not urine protein quantification [9]. In the “Izumo City CKD Exacerbation Prevention Countermeasures” program, urine protein quantification is to be performed when re-examination candidates are tested by their family doctors.

Letter of invitation to be sent to a re-examination respondent “who fulfilled the following conditions: estimated glomerular filtration rate (eGFR), <50 mL/min/1.73 m^2^ (eGFR < 40 mL/min/1.73 m^2^ for those aged > 70 years), serum Cr, >1.01 mg/dL for males and >0.9 mg/dL for females; and urinary protein, >2+”. 

A request form was sent to nephrology specialized institutions when family doctors conducted a re-examination and determined that detailed examinations were necessary. There are five copies: referral source copy, copy from the referral source to Izumo City, copy from the referral source to the nephrologist, copy from the nephrologist to the referral source, and copy from the nephrologist to Izumo City.

We compared the rates of CKD re-examination candidates, re-examinations, nephrologist referrals, and city health guidance referrals between FY2018 and FY2019. The survey data in this study were provided by the Izumo City Office in consideration of not identifying personal information.

The baseline characteristics stratified according to fiscal year were compared using the unpaired t-test, the Wilcoxon rank-sum test, or χ^2^ test, as appropriate. The number of CKD re-examination candidates, number and rate of re-examinations, and number and rate of nephrologist referrals were analyzed using the χ^2^ test. Fisher’s exact test was used to analyze the number and rate of city health guidance referrals. Statistical significance was set at *p* < 0.05. Stata/SE 16.0, StataCorp LLC, College Station, TX, USA) was used for the statistical analysis.

The study protocol was approved by the Ethics Committee of the Shimane University Faculty of Medicine (No. 20210625-3). The requirement for informed consent from all the participants was waived, according to the Japanese Ethical Guidelines for Medical and Health Research Involving Human Subjects. 

## 3. Results

The baseline characteristics of the adults who underwent specific health checkups in FY 2018 were similar to those in FY 2019 (Appendix A). Additionally, the baseline characteristics of the CKD re-examination candidates (Appendix A) and CKD re-examination implementation (Appendix A), were comparable between FY2018 and FY2019.

The number of patients who underwent specific health checkups in FY2018 was 11,685. Among those, the CKD re-examination candidates were 869 (7.4%). Among the re-examination candidates, 255 underwent re-examinations, 207 did not, and the status was unknown for 407. Among the patients who underwent re-examination, 20 were referred to nephrologists and one underwent health guidance.

The number of patients who underwent specific health checkups in FY2019 was 11,315, and 906 (8.0%) of them were CKD re-examination candidates. Among the re-examination candidates, 428 underwent re-examination, 174 did not, and the status was unknown for 304. Among the re-examined patients, 35 were referred to the nephrologists and 31 underwent health guidance.

The CKD re-examination candidate rate was ~ 7% in both years and showed no significant differences (*p* = 0.105) (Table 1). 

Significant differences were observed in the re-examination implementation, non-implementation, and unknown status rates between FY2018 and FY2019, and re-examination implementation rate increased significantly (*p* < 0.001) (Table 2). Even investigations that excluded the subjects with unknown status showed significant increase in the re-examination implementation rate (*p* < 0.001) (data not shown).

Subsequent trends of re-examination candidates showed increase in the number of nephrologist referrals; however, the difference was not significant (*p* = 0.877) (Table 3). 

However, the rate of health guidance referrals significantly increased in FY2019 (*p* < 0.001) (Table 4).

## 4. Discussion

To diagnose CKD at an early stage, it is important to promote appropriate health guidance and consultation recommendations such as regular medical examinations and implement continuous high-quality and appropriate treatment. From FY 2018, Izumo City started the “Izumo City CKD Exacerbation Countermeasures” program. In this study, we aimed to report the methods undertaken and the effects of this program.

CKD has only a few symptoms, and it is often already advanced by the time symptoms appear. It is difficult to cure exacerbated kidney diseases and may lead to end-stage renal failure and cardiovascular diseases. Therefore, early detection and treatment are important to prevent, exacerbation of the disease. Thus, the “Kidney Disease Countermeasures Study Group” investigated ways to implement kidney disease countermeasures in 2008 and compiled their results in the “Future Kidney Disease Countermeasures” [10]. This report outlined the objectives of “preventing the exacerbation of renal dysfunction and progression to dialysis due to chronic renal failure” and “suppressing the onset of cardiovascular diseases (e.g., cerebrovascular diseases, and myocardial infarction) associated with CKD.” Various countermeasures used over the past 10 years have resulted in some degree of success [11]; however, the number of CKD patients continue to increase. Therefore, the “Kidney Disease Countermeasures Study Group Report: Aiming for Further Promotion of Kidney Disease Countermeasures” was issued in 2018 to further promote kidney disease countermeasures [4,5]. The objective of this report had the objective “to thoroughly prevent CKD exacerbation, maintain, and improve the quality of life of patients with CKD (including dialysis and kidney transplantation patients). In these patients, CKD should be diagnosed at an early stage and high-quality and appropriate treatment should be provided.” To that end, this report has suggested the implementation of the following initiatives in the future: “dissemination and public awareness,” “improvement of local medical care provision systems,” “improvement of medical care standards,” “human resource development,” and “promotion of research and development.” Furthermore, through this program, the goal is to reduce the annual number of new dialysis patients to <35,000 by 2028.

Additionally, specific health checkups [12] have been implemented in Japan since 2008. The focus has been on metabolic syndrome with the aim of preventing the onset and exacerbation of lifestyle-related diseases, such as diabetes, hyperlipidemia, and hyperuri cemia. These checkups are conducted in order to accurately identify those who need specific health guidance to reduce the number of individuals who may suffer from lifestyle-related diseases in the future.

Urinalysis was initially set as a specific health checkup item, though serum Cr was excluded. However, considering the increase in the number of dialysis patients with diabetes and nephropathy as underlying diseases, serum Cr levels have also been measured at each municipal level.

The Japan Kidney Association [13] was established in Japan in February 2018 prior to the issuance of the “Kidney Disease Countermeasures Study Group Report”. Furthermore, the Health, Labor, and Welfare Policy Research Grant (Kidney Disease Policy Research Project) funded the studies, “Progress management of countermeasures based on the Kidney Disease Countermeasures Study Group Report and the establishment of evidence that contributes to proposals for new measures” and “Promotion of nationwide dissemination and public awareness of CKD and contributions to medical care through the establishment of a local medical cooperation system” research group [14] to establish CKD countermeasures on a national level [15].

CKD countermeasures were initiated throughout Japan, though this was not possible with just medical professionals such as family doctors or nephrologists; it was essential to have the cooperation of government agencies. Some municipalities independently conducted CKD countermeasures; however, many of them conducted those with the aim of preventing the exacerbation of diabetic nephropathy. In Izumo City, the Izumo Area Diabetes Countermeasures Study Group has been conducting activities with the cooperation of the government, Izumo Medical Associations, diabetes specialists, dental associations, and pharmaceutical associations, with nephrologists being involved since 2014. With these efforts, awareness of CKD countermeasures has steadily increased. The “Izumo City CKD Exacerbation Countermeasures” began in 2018.

Based on repeated discussions by study groups, it was decided that re-examinations should be recommended to those who meet the following conditions: “eGFR < 50 mL/min/1.73 m^2^ (eGFR < 40 mL/min/1.73 m^2^ for those aged >70 years), serum Cr, >1.01 mg/dL for males and >0.9 mg/dL for females, and urinary protein >2+.” Approximately 7% of those who underwent medical examinations met these criteria in both FY2018 and FY2019. The re-examination implementation rate was 26.8% in FY2018, which significantly increased to 43.9% in FY2019, although this was thought to be due to increased awareness of CKD among the public. Since FY2011, we have launched the Shimane Prefecture Chronic Kidney Disease Countermeasures Council and held events twice a year. On World Kidney Day in March, we gave out handouts to raise public awareness of CKD and conducted health consultations in a commercial department store in Izumo City. Half a year later, in August–September, we hosted a public lecture (speakers included physicians from various clinical departments such as nephrology, cardiology, and diabetology, pharmacists, registered dietitians, and nurses qualified for kidney disease treatment). We believe that these activities have gradually increased the public awareness of CKD. Furthermore, it is thought that there is an increased enthusiasm toward the dissemination and public awareness of CKD among the government and nephrologists after the launch of the “Izumo City CKD Exacerbation Prevention Countermeasures” program.

However, nephrologist referrals did not increase after the implementation of re-examination. Several possible factors can be considered for this observation. One factor is the possibility that no abnormalities were observed during the re-examination. Patients often refrain from food and water intake the night before undergoing a health examination. This can concentrate the urine and increase the risk of urinary protein positivity; however, improvements were seen as a result of proper water intake at the time of re-examination. Another factor is the possibility that the patients did not wish to visit a university hospital or core hospital, which are nephrology specialized institutions, even if abnormal values were present, since no subjective symptoms were present for CKD. Furthermore, family doctors in Izumo City have long been actively participating in CKD-related lectures, and there is the possibility that they have been able to conduct follow-up examinations to the extent that they can examine patients in their own hospital even after re-examination, with referrals given upon exacerbation. In fact, the number of health guidance referrals in Izumo City has greatly increased; hence, it may be considered that family doctors are actively participating in CKD medical care.

The rate of re-examination non-implementation decreased in FY2019; however, the rate of patients undergoing regular observations by family doctors remained unchanged at 70%, and the rate of patients undergoing treatment by nephrologists increased (data not shown). The data in this study could not clarify whether those who were referred to nephrologists in FY2018 were in the process of being treated by nephrologists in FY2019. Furthermore, the number of unexamined patients decreased even when re-examinations were not implemented; hence, it was considered that several patients might have gone to medical institutions in some way and that early detection can be expected in the future.

The limitations of this study are that it was a retrospective study and no investigations on changes in renal function or the presence of dialysis were conducted. Future work will require investigations on data collection and tracking methods in collaboration with Izumo City to verify the effectiveness of CKD countermeasures. Moreover, increase in the number of follow-up years would allow further investigation of the changes in eGFR and the development of end-stage kidney disease. Further, although the number cases with unknown status has decreased from FY2018 to FY2019, the proportion still remains high; therefore, this may have affected the results of this program. We are currently working with the Izumo City staff to find a way to reduce the number of cases with unknown status.

Additionally, the current referral criteria created by the Ministry of Health, Labor, and Welfare [16] and the Japanese Society of Nephrology [17] stipulate “referrals for those with urinary protein >1+” As a result, it is possible that conducting comparative studies at a national level may be difficult, and Izumo City is also investigating options to revise their re-examination criteria to the national criteria in the future.

## 5. Conclusions

We launched the “Izumo City CKD Exacerbation Prevention Countermeasures” as part of a program for nationwide kidney disease countermeasures. This has not reached the key performance indicator of the Kidney Disease Countermeasures Study Group Report of decreasing the number of patients introduced to dialysis; however, increased re-examination and health guidance examination rates indicating increased awareness of CKD among the public and family doctors were observed. We anticipate further results in the future, and will continue to strive for dissemination and public awareness, improvement of local medical care provision systems, improvement of medical care standards, and human resource development. 

## Figures and Tables

**Figure 1 jpm-11-01104-f001:**
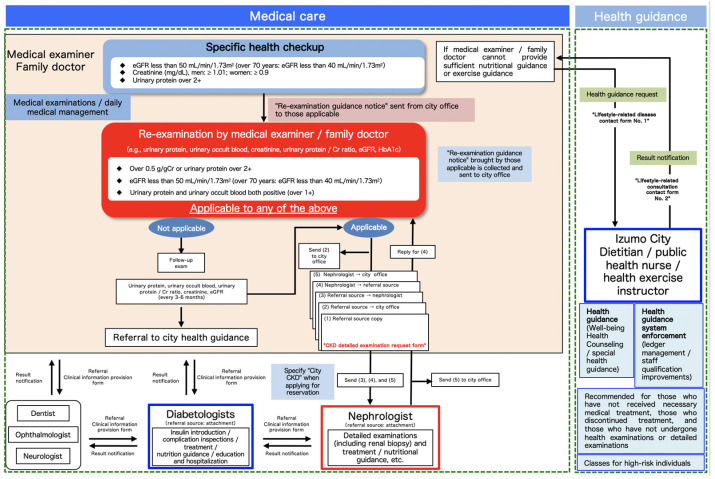
Izumo City CKD Exacerbation Prevention Countermeasures flow chart showing specific health checkup and family doctor CKD determination criteria. The flow chart shows subsequent referrals to nephrologists/diabetes specialists, dentists, ophthalmologists, neurologists, or well-being health counseling (health guidance). CKD, chronic kidney disease; Cr, creatinine; and eGFR, estimated glomerular filtration rate.

**Figure 2 jpm-11-01104-f002:**
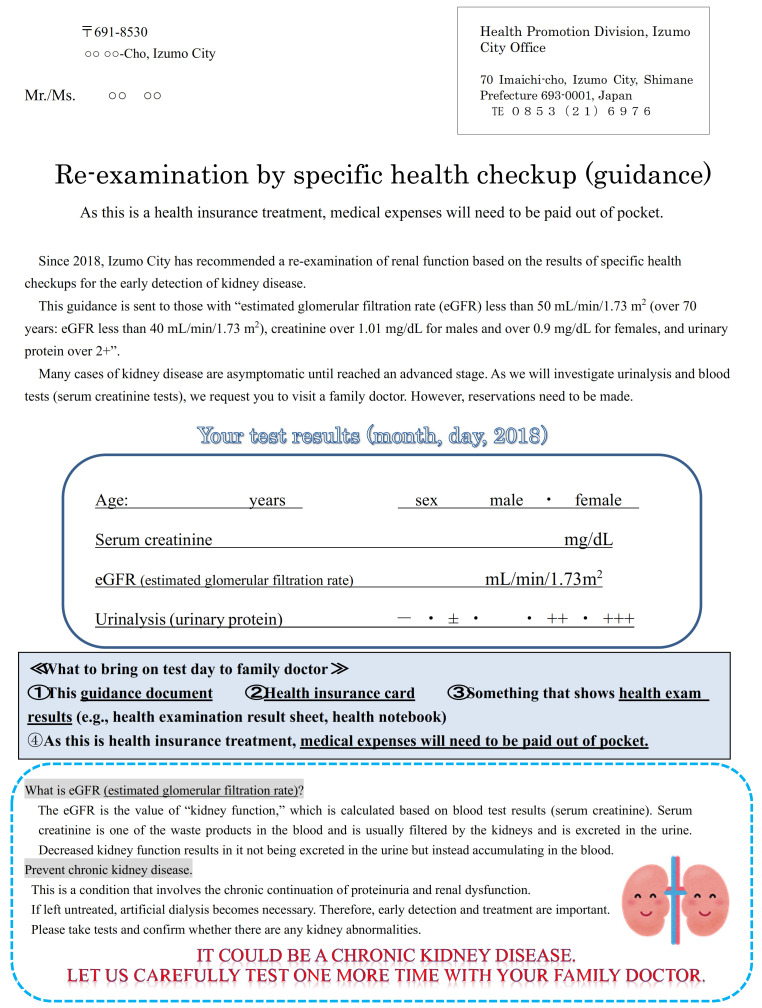
Re-examination through specific health checkups (guidance).

**Figure 3 jpm-11-01104-f003:**
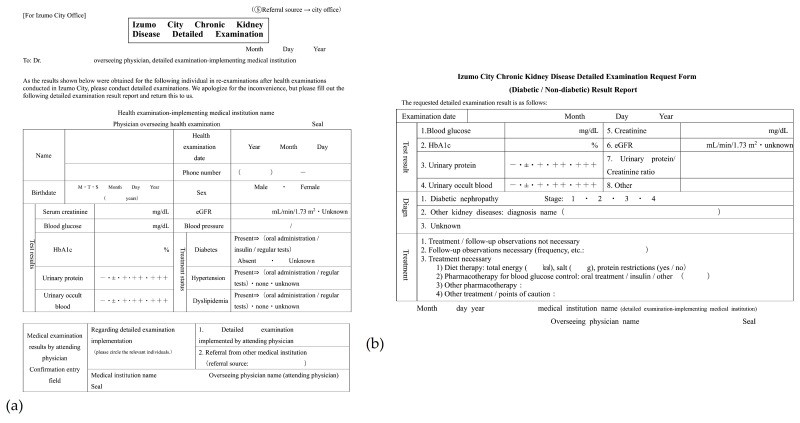
Izumo City Chronic Kidney Disease Detailed Examination Request Form. (**a**) Detailed Examination Request Form; (**b**) Detailed Examination Result Report.

**Table 1 jpm-11-01104-t001:** Rate of CKD re-examination candidates.

	FY2018	FY2019	*p*-Value ^1^
Number	11,685	11,315	
Re-examination candidates, number (%)	869 (7.4)	906 (8.0)	0.105
Re-examination non-candidates, number (%)	10,816 (92.6)	10,409 (92.0)	

^1^*p*-value was calculated using χ^2^-test; *p* = 0.105. CKD, chronic kidney disease; and FY, fiscal year.

**Table 2 jpm-11-01104-t002:** CKD re-examination implementation rate among re-examination candidates.

	FY2018	FY2019	*p*-Value ^1^
Number	869	906	
Re-examination implementation, number (%)	255 (29.4)	428 (47.2)	<0.001
Re-examination non-implementation, number (%)	207 (23.8)	174 (19.2)	
Unknown status, number (%)	407 (46.8)	304 (33.6)	

^1^*p*-value was calculated using χ^2^-test; *p* < 0.001. CKD, chronic kidney disease; FY, fiscal year.

**Table 3 jpm-11-01104-t003:** Rate of nephrologist referrals for individuals undergoing re-examination.

	FY2018	FY2019	*p*-Value ^1^
Number	255	428	
Nephrologist referrals, number (%)	20 (7.8)	34 (7.9)	0.962
Follow-up at their own hospital, number (%)	235 (92.2)	394 (92.1)	

^1^*p*-value was calculated using χ^2^-test; *p* = 0.962. FY, fiscal year.

**Table 4 jpm-11-01104-t004:** Health guidance referrals for patients undergoing follow-ups at their own hospital.

	FY2018	FY2019	*p*-Value ^1^
Number	235	394	
Health guidance referrals, number (%)	1 (0.4)	31 (7.9)	<0.001
No referral, number (%)	234 (99.6)	363 (92.1)	

^1^*p*-value was calculated using Fisher’s exact test.

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
