# Peer review of "Effectiveness of CKD Exacerbation Countermeasures in Izumo City"

_jpm, 2021, doi:10.3390/jpm11111104_

Round 1
Reviewer 1 Report
Nothing of new, but interesting and well done.
More of these programs should be implemented around the world.
Author Response
We appreciate your insightful comments on our manuscript.
Reviewer 2 Report
Ito et al. reported the results of the japanese program for CKD exacerbation in the city of Izumo.
Whilst the aims of this project are of concern, the rate of CKD re-examination seems particularily low, possibly because of the eGFR level < to 50ml/min (stage II, near stage IIIa).
It would have been of interest to know the eGFR evolution of patients participating to this health program, i.e. if this permitted to slow down evolution towards ESRD.
Finally, a large number of patients had an unknown status (near 50%), which can be consider as a bias or a limit of this health prevention program.
Author Response
We appreciate your helpful comments on our manuscript. Based on your suggestions, we have made the following revisions in our manuscript. We hope that the quality of our manuscript has been improved adequately.
- Ito et al. reported the results of the Japanese program for CKD exacerbation in the city of Izumo. Whilst the aims of this project are of concern, the rate of CKD re-examination seems particularily low, possibly because of the eGFR level < to 50ml/min (stage II, near stage IIIa).
We appreciate this insightful comment on our manuscript. As pointed out by the reviewer, the criterion used in this program was eGFR level <50 mL/min; accordingly, the rate of CKD re-examination might have been low. However, according to the CKD Clinical Practice Guide 2012, which is followed in Japan, patients with eGFR level <50 mL/min should be referred to a nephrologist. Therefore, we used this criterion in this program.
We have added the following sentence in the Introduction:
“(Page 2, lines 61-65) According to the Clinical Practice Guidebook for Diagnosis and Treatment of Chronic Kidney Disease 2012 [7], which is followed in Japan, patients with an estimated glomerular filtration rate (eGFR) level <50 mL/min should be referred to a nephrologist; we used this criterion for patient selection in this program.”
- It would have been of interest to know the eGFR evolution of patients participating to this health program, i.e. if this permitted to slow down evolution towards ESRD.
As pointed out by the reviewer, we would like to further study the changes in eGFR and the development of end-stage kidney disease following an increase in the number of follow-up years.
We have added the following sentence in the Discussion:
“(Page 7, lines 294-295) Moreover, increase in the number of follow-up years would allow further investigation of the changes in eGFR and the development of end-stage kidney disease.”
- Finally, a large number of patients had an unknown status (near 50%), which can be consider as a bias or a limit of this health prevention program.
Although the number of cases with unknown status has decreased from FY2018 to FY2019, the proportion still remains high, and as pointed out by the reviewer, it cannot be denied that this might have affected the results of this program.
We are currently working with the Izumo City staff to find a way to reduce the number of cases with unknown status.
We have added the following sentences in the Discussion:
“(Page 8-9, lines 295-299) Further, although the number of cases with unknown status has decreased from FY2018 to FY2019, the proportion still remains high; therefore, this may have affected the results of this program. We are currently working with the Izumo City staff to find a way to reduce the number of cases with unknown status.”
Reviewer 3 Report
Concerning reviewed paper I have some serious notes.
- Lack of information about Ethical Committee Agreement.
- The health check-up plan was complicated, including receiving a referral for a follow-up check-up.
- During the qualification of patients , proteinuria was only determined besed on urine test, this result was not verified, by UPCR.
- The method of calculation of eGFR in the study population was not given.
- The details of characteristics of the study population was not given in the paper.
- There was no information about comorbidity based on the Charlson index, the frailty scale, nutrition status of patients and pharmacological treatment. All these factors influence on eGFR or proteinuria.
- No exclusion criteria were presented in the study.
- Were there qualified patients with malignancies, severe cardiovascular disease, cachexia?
- The implementation of the proposed health check-up plan for detection the early stage of CKD is useless for everyday clinical practice. This flowchart should be simplified for future everyday clinical practice.
Author Response
We appreciate your helpful comments on our manuscript. Based on your suggestions, we have made revisions in our manuscript as detailed below. We wish that the quality of our manuscript has improved adequately.
Concerning reviewed paper I have some serious notes.
1. Lack of information about Ethical Committee Agreement.
Thank you for pointing this out to us.
We added the following sentences in the Methods section:
“(Page 4, lines 139-142) The study protocol was approved by the Ethics Committee of the Shimane University Faculty of Medicine (No. 20210625-3). The requirement for informed consent from all participants was waived, according to the Japanese Ethical Guidelines for Medical and Health Research Involving Human Subjects.”
2. The health check-up plan was complicated, including receiving a referral for a follow-up check-up.
We appreciate the critical comment regarding this aspect of our study. In Japan, we do not have a large-scale health checkup system for CKD. Therefore, we created a program aimed at CKD prevention by using the specific health checkup system focused on prevention of metabolic syndromes. In this program, participant eligibility for CKD re-examination was identified according to the Clinical Practice Guidebook for Diagnosis and Treatment of Chronic Kidney Disease 2012.
We plan to consider program modifications in the future based on the problems associated with the current operation.
3. During the qualification of patients, proteinuria was only determined based on urine test, this result was not verified, by UPCR.
We appreciate the critical comment regarding this aspect of our study.
In the specific health checkup in Japan, urine qualitative analysis is one of the required items, and urine protein quantification is not a required item [9]. In this "Izumo City CKD Exacerbation Prevention Countermeasures", urine protein quantification is to be performed when re-examination candidate is tested by their family doctors.
We added the following sentences in the Methods as follows:
“(Page 3, lines 102-105) In Japan, the specific health checkup includes urine qualitative analysis as one of the required items, but not urine protein quantification [9]. In the “Izumo City CKD Exacerbation Prevention Countermeasures” program, urine protein quantification is to be performed when re-examination candidates are tested by their family doctors.”
4. The method of calculation of eGFR in the study population was not given.
Thank you for pointing this out to us.
We added the following sentences in the Methods section:
“(Page 3, lines 98-101) The eGFR was calculated at each health checkup facility using a three-variable equation modified for Japanese patients [8]:
eGFR (mL/min/1.73 m2) = 194 × age (years)-0.287× serum creatinine (mg/dL)-1.094× 0.739 (if female) .”
5. The details of characteristics of the study population was not given in the paper.
We appreciate this comment regarding our manuscript.
The details of the baseline characteristics of the adults who underwent specific health checkups in FY 2018 and FY 2019 are shown in Table S1. Although there was a statistically significant difference between the two groups in some items (age, systolic blood pressure, triglyceride, low-density lipoprotein cholesterol, hemoglobin A1c, and eGFR), the actual values were almost the same; therefore, we considered there to be no significant difference between the groups. The details of the baseline characteristics of the CKD re-examination candidates and CKD re-examination implementation in FY 2018 and FY 2019 are shown in Tables S2 and S3. There was no significant difference in the baseline characteristics between the two groups.
We have added the following sentences in the Methods and Results sections:
“(Page 4, lines 132-133) The baseline characteristics stratified according to the fiscal year were compared using the unpaired t-test, the Wilcoxon rank-sum test, or the χ2, as appropriate.”
“(Page 4, lines 146-149) The baseline characteristics of the adults who underwent specific health checkups in FY 2018 were similar to those in FY 2019 (Table S1). Additionally, the baseline characteristics of the CKD re-examination candidates (Table S2) and CKD re-examination implementation (Table S3) were comparable between FY2018 and FY2019.”
6. There was no information about comorbidity based on the Charlson index, the frailty scale, nutrition status of patients and pharmacological treatment. All these factors influence on eGFR or proteinuria.
We appreciate this critical comment regarding our manuscript.
It was difficult to obtain information regarding the various factors influencing eGFR or proteinuria that have been pointed out by the reviewer, based on the results of the specific health checkups; however, we did obtain information regarding hypertension, diabetes, dyslipidemia, and CVD. A comparison of these factors in the CKD re-examination candidates showed no statistically significant difference between FY 2018 and FY 2019 (Table S2).
7. No exclusion criteria were presented in the study.
Thank you for pointing this out to us.
In this study, all candidates eligible for CKD re-examination were included.
We have added the following sentences in the Methods section:
“(Page 2-3, lines 78-82) Subjects aged 40-74 years who underwent the Izumo City National Health Insurance special health checkup in the fiscal year (FY) 2018 (11,685 people) and FY2019 (11,315 people) were included in the study. There were 869 individuals who met the re-examination criteria in FY2018 and 906 in FY2019, all of whom were included in the analysis.”
8. Were there qualified patients with malignancies, severe cardiovascular disease, cachexia?
Thank you for your question.
The purpose of the specific health checkup in Japan is to screen for metabolic syndromes, and I believe that it plays a role in the prevention of CVD. However, patients with advanced cancer or cachexia are generally not subjected to the health screening, and it is unlikely that this study will be able to collect this information.
9. The implementation of the proposed health check-up plan for detection the early stage of CKD is useless for everyday clinical practice. This flowchart should be simplified for future everyday clinical practice.
Thank you for your suggestion.
The specific health checkup system has been implemented in Japan since 2008, and the focus has been on metabolic syndromes with the aim of preventing the onset and exacerbation of lifestyle-related diseases, such as diabetes, hyperlipidemia, and hyperuricemia.
Since numerous people in Japan undergo this specific health checkup, and metabolic syndromes are strongly associated with CKD, we have developed a screening system for CKD using this specific health checkup. For the reasons mentioned above, it is undeniable that the current system is somewhat complicated; however, we hope to upgrade the system in the future to increase the rate of CKD re-examination candidates and health guidance referrals.
Round 2
Reviewer 2 Report
Thank you for your answers.
Despite data about eGFR evolution on included patients are not shown, the discussion improved and some limits are now explicited to the readers.
Reviewer 3 Report
I have received comprehensive answers to all questions from
the previous review. I have no new comments.